# In-Silico Analysis of pH-Dependent Liquid-Liquid Phase Separation in Intrinsically Disordered Proteins

**DOI:** 10.3390/biom12070974

**Published:** 2022-07-12

**Authors:** Carlos Pintado-Grima, Oriol Bárcenas, Salvador Ventura

**Affiliations:** Institut de Biotecnologia i Biomedicina, Departament de Bioquímica i Biologia Molecular, Universitat Autònoma de Barcelona, Bellaterra, 08193 Barcelona, Spain; carlos.pintado@uab.cat (C.P.-G.); oriol.barcenas@autonoma.cat (O.B.)

**Keywords:** pH, liquid-liquid phase separation, intrinsically disordered proteins, protein solubility, protein disorder, mutations, bioinformatics

## Abstract

Intrinsically disordered proteins (IDPs) are essential players in the assembly of biomolecular condensates during liquid–liquid phase separation (LLPS). Disordered regions (IDRs) are significantly exposed to the solvent and, therefore, highly influenced by fluctuations in the microenvironment. Extrinsic factors, such as pH, modify the solubility and disorder state of IDPs, which in turn may impact the formation of liquid condensates. However, little attention has been paid to how the solution pH influences LLPS, despite knowing that this process is context-dependent. Here, we have conducted a large-scale in-silico analysis of pH-dependent solubility and disorder in IDRs known to be involved in LLPS (LLPS-DRs). We found that LLPS-DRs present maximum solubility around physiological pH, where LLPS often occurs, and identified significant differences in solubility and disorder between proteins that can phase-separate by themselves or those that require a partner. We also analyzed the effect of mutations in the resulting solubility profiles of LLPS-DRs and discussed how, as a general trend, LLPS-DRs display physicochemical properties that permit their LLPS at physiologically relevant pHs.

## 1. Introduction

Our view of cellular organization has been progressively changing over the last decade. Since the discovery of the first liquid droplets in *C. elegans* embryo’s cells [1], numerous membrane-less organelles (MLOs) with a wide variety of biological functions have been described in different organisms [2,3,4,5]. In contrast to their classic membrane-enclosed counterparts, MLOs are dynamic supramolecular structures that can undergo reversible liquid–liquid phase separation (LLPS) in response to specific stimuli [6,7]. The reversible and tunable nature of LLPS has turned into an effective compartmentalization mechanism for the always-changing cellular milieu, allowing the selective spatiotemporal formation of biomolecular condensates [8]. MLOs are often enriched in intrinsically disordered proteins (IDPs) and/or proteins bearing unstructured regions and low-complexity domains [9,10,11]. These sequences play an important role in droplet formation by establishing multivalent weak intermolecular interactions [12,13]. Indeed, mutations in some of these unstructured regions might deregulate the equilibrium of LLPS and lead to the onset of neurodegenerative diseases such as amyotrophic lateral sclerosis (ALS), frontotemporal dementia (FTD) [10,14,15], or muscular dystrophies [16,17].

Intrinsically disordered regions (IDRs) are exposed to the solvent and thus influenced by the protein microenvironment, whose fluctuation can trigger conformational switches [18]. This context-dependency also applies to LLPS processes, and parameters such as ionic strength, temperature, or pH have been described as regulatory elements in these reactions [6]. Other factors apart from solvent conditions can also influence the behavior of IDPs in LLPS, including the protein concentration and the presence of other biomolecules, mainly RNA and partner proteins [19]. Therefore, the capacity of a protein to phase-separate is not binary and cannot be univocally attributed to the intrinsic properties of the sequence as it strongly depends on the specific conditions of the cellular milieu at any given time.

In recent work, we accounted for the effect of pH on the solubility [20] and disorder state [21] of IDPs using different equations that simultaneously consider the impact of the pH on sequence hydrophobicity and net charge. This allowed us to develop two novel bioinformatic tools to predict pH-dependent solubility and disorder: SolupHred [22] and DispHScan [23], respectively, which recapitulate previous and novel experimental data [24,25,26]. The solubility and degree of disorder of a protein at given pH influence its propensity to phase-separate, but large-scale analyses exploring these connections are scarce. To provide insights on the role of pH in LLPS, we conducted an in-silico study of solubility and disorder at different pH values for LLPS-involved disordered regions (LLPS-DRs), accounting for a total of 1600 sequences, using the SolupHred and DispHScan algorithms.

## 2. Materials and Methods

### 2.1. Dataset Generation

LLPS regions were extracted from PhaSepDB, a manually curated database of liquid-liquid phase separation-related proteins and MLOs [27]. Afterward, the disordered nature of these regions was surveyed using the IUPRED2A server [28]. High-confidence LLPS-DRs (sequences longer than 20 amino acids with an IUPRED score ≥ 0.5) were saved for the analyses (*n* = 1600). Two different subgroups were created according to the ability of each LLPS region to phase-separate by itself (psself) or with the help of a partner (psother). A second dataset of general disordered segments was generated for comparison purposes. IDRs longer than 20 amino acids were obtained from DisProt (release 2021_08), a manually curated database of experimentally validated IDPs [29].

### 2.2. pH-Dependent Solubility and Disorder Analyses

The effect of pH on both LLPS-DRs’ solubility and disorder was predicted by two in-house algorithms considering this key variable in their pipelines. SolupHred and DispHScan web servers recalculate protein lipophilicity and net charge as a pH function to predict IDPs’ solubility and disorder in a specific pH context [22,23]. All disordered sequences were run with SolupHred and DispHScan in a pH range between 4 and 9, using a step size of 0.1 to account for small variations. The software output solubility and disorder scores, respectively. The disorder score is named the DispH score, with lower values indicating a more disordered state. Statistical significance between variables and/or datasets was assessed with Mann-Whitney-Wilcoxon two-sided test with Bonferroni correction. *p*-values were marked with asterisks to better convey statistical significance (*p* > 0.05 (ns), *p* ≤ 0.05 (*), *p* ≤ 0.01 (**), *p* ≤ 0.001 (***), *p* ≤ 0.0001 (****)). As for the statistical significance of the linear regression (whether the distribution follows a slope of 0 or not), a Wald Test with t-distribution of the test statistic was used. Statistical tests are described in the Appendix A of this article.

## 3. Results

### 3.1. LLPS-DRs Present Maximum Solubility around Neutral pH

One of the main uncertainties when dealing with IDPs and LLPS is the conformational state that disordered segments present during the phase-separation process. Some studies suggest that LLPS-DRs contain aggregation-prone regions (APRs), which are needed for condensate formation, and that they act by reducing the solubility in the conditions in which LLPS occurs [30,31]. To explore this hypothesis, we analyzed the pH-dependent solubility of LLPS-DRs in the 4 to 9 pH range, comparing the obtained results with the pHs at which LLPS was described. Given that similar solubility scores may span various pH units, we assumed 10% maximum/minimum solubilities as the most representative variables for the analysis. The results revealed that solubility in LLPS-DRs tends to be maximum around neutral pH (μ = 6.96, σ = 1.29, Figure 1A), whereas minimum solubilities are mainly achieved at the extremes of the analyzed pH interval (Figure 1B). When studying buffer conditions described for LLPS, a similar result was observed (*n* = 181, μ = 7.50; σ = 0.43). This suggests that LLPS-DRs are soluble near physiological pH values and that LLPS occurs at pHs at which, as a group, these protein regions display significant solubility. To further support these findings, the absolute differences in maximum and minimum solubility scores were individually compared with those obtained at the pH where LLPS was described (Figure 2A). The analysis confirms that, as a trend, effective LLPS-promoting pHs resemble more those at which the correspondent proteins are maximally soluble than those with minimal solubility.

Overall, the results indicate that pH solution conditions favoring high solubility overlap with those at which condensation is experimentally observed, suggesting that these molecular signatures might be important for LLPS and relevant at the cellular physiological pH. This is surprising since it seems to contradict the general assumption that pro-aggregational conditions are necessary for LLPS.

### 3.2. A Link between pH-Dependent Solubility and pH-Dependent Disorder in LLPS-DRs

We performed the same kind of analysis as above to assess whether pHs at which LLPS is reported coincide with conditions that favor disorder or, on the contrary, promote compactness of LLPS-DRs (Figure 2B). The data indicate that minimum DispH scores and thus larger disorder content match better conditions at which LLPS experimentally occurs.

The relationship between solubility and conformational disorder in the LLPS-DR dataset was then analyzed. When plotting the solubility and disorder scores at neutral pH (pH = 7.0), a highly significant linear correlation was observed between these two variables (R^2^ = 0.90, *p* < 1^−10^) (Figure 3), suggesting that solubility and disorder are intimately ligated in this dataset. Furthermore, higher solubility scores were correlated with minimum DispH scores (maximum disorder) and vice-versa. This indicates that regions that populate a more soluble sequence space are also more prone to be disordered. This makes sense since the presence of low-soluble sequences with a high degree of disorder would render them aggregation-prone and thus harmful at the analyzed physiological pH, whereas a higher degree of compactness would protect their hydrophobic residues from exposure to the solvent, at least transiently. Conversely, highly soluble sequences would find it challenging to attain a compact conformation, and their presence in an unfolded state would not represent a significant risk of establishing aberrant hydrophobic interactions with other cellular components.

### 3.3. Different Datasets in LLPS-DRs Present Distinct Property Distributions

Not all proteins present in MLOs can undergo independent LLPS [19]. As we said, this process is highly context-dependent [6], and many proteins require a partner, often a nucleic acid molecule, to phase-separate. Therefore, the properties of LLPS-DRs may vary according to their ability to undergo LLPS by themselves (psself) or with a partner (psother). Indeed, no differences in calculated solubility and disorder at pH 7.0 were observed between the entire LLPS-DRs dataset (without differentiating psself and psother sequences) and the IDRs present in DisProt. Conversely, when the study considered psself and psother regions separately, the significance level dramatically increased when compared with DisProt and between them. This indicates that psself and psother sequences need to be studied independently.

Psself regions showed lower solubility scores than psother and DisProt (Figure 4A), which, not surprisingly, was associated with higher DispH scores (more ordered) (Figure 4B). These differences are more apparent in the distribution plot of solubility and disorder scores for psself and psother datasets at pH 7.0 (Figure 4C,D). This indicates that psself regions tend to be less soluble and disordered than psother at pH 7.0. This allows us to reconcile our analysis with the view that aggregation propensity is a player in LLPS and to reformulate it. For psself proteins, which can separate efficiently and autonomously, the situation is not that they experiment LLPS at pHs at which their solubility is at a minimum, but instead that its intrinsic solubility at physiological pH, despite significant, is below the average solubility of disordered proteins, which facilitate homotypic self-assembly. An inverse trend applies to disorder. Still, this association only applies for psself proteins since proteins that require an interactor to be incorporated in MLOs are not only more soluble than psself ones, but also that the conjunct of IDRs in DisProt and will find difficulties to self-assemble without a scaffolding molecule. All these biophysical connections are masked when studying the LLPS dataset as a homogeneous protein group.

### 3.4. Psself Regions Present Lower Dispersion in Solubility and Disorder in the Physiological pH Range

When the differences between maximum and minimum solubility and disorder scores were analyzed in the entire selected pH range (from pH 4.0 to pH 9.0 with a step size of 0.1, as detailed in the Section 2), DisProt exhibited the broader dispersion of both solubility and disorder parameters, with psself being the dataset with the lowest dispersion (Figure 5A,B). This suggests that psself regions are less sensible to pH fluctuations and populate a narrower solubility-disorder space, likely compatible with cellular conditions in which LLPS can still occur even if pH deviations arise, as long as they are not very large. On the other hand, the dispersion of psother regions is significantly wider for both parameters, suggesting a lower selective pressure to keep their solubility/disorder properties restricted in a cell-compatible pH gradient, which is expected, as they are not competent for phase separation alone.

### 3.5. Case Study of Independent LLPS Happening at Physiological pH

To contextualize the previous results with defined cases of LLPS, we sought the literature for IDPs whose LLPS was experimentally demonstrated to occur at physiological pH.

An example of a well-characterized LLPS-DR is the low complexity domain (LCD) of TDP-43, a protein associated with neurodegenerative disorders such as ALS or FTD [10,14,32]. Experimental studies indicate that LLPS of the TDP-43 LCD is pH-dependent [33]. LLPS was tested at three different pH values (4.0, 6.0, and 7.0) with different salt concentrations (from 0 to 300 mM NaCl). The presence of liquid droplets was observed in all salt conditions at pH 7.0 and progressively dissipated as the pH decreased, requiring higher concentrations of salt to observe LLPS. These results reveal that physiological conditions are conductive of LLPS for the TDP-43 LCD. When analyzing the resulting solubility curve predicted by SolupHred for this LLPS-DR in this pH range (Figure 6A), maximum solubility is achieved at pH 7.0. The LCDs of the hnRNPA1 [34] or the U1-70K proteins [35] have been shown to form liquid droplets at pH 7.0, where the solubility score is close to the maximum (Figure 6B,C). Importantly, dysfunctional LLPS of these proteins lead to aberrant aggregation and neurodegeneration.

Altogether, for these proteins, LLPS occurs in a high solubility range that is compatible with physiological conditions. This does not imply that LLPS necessarily occurs more efficiently at a pH in which protein solubility is strictly at its maximum, but rather that to form biomolecular condensates, proteins should be in a solubility regime that enables them to diffuse and interact homotypically and, if needed, with their partners. Conditions of very low solubility can also promote LLPS, but they are often associated with aggregation and pathogenicity.

### 3.6. Mutations in LLPS Formation and Disease

Mutations in LLPS-DRs can hinder the capacity of proteins bearing these sequences to phase-separate. This has been studied in vitro, analyzing the formation of liquid condensates using different variants of a given sequence. Unveiling how these mutations affect the solubility pattern of these regions is vital to understanding their connection with LLPS deregulation.

It is well established that tyrosines are important residues in LLPS since their involvement in π-π [36] or cation-π [11,37] interactions are key for the weak multivalent contact that sustain liquid droplets. In this way, an increasing number of Tyr-to-Ser substitutions in the LCD of FUS has been linked with a reduction in the formation of droplets at pH 7.5 [9]. Therefore, we plotted the relative fluorescence intensity after 10 min, a measure of LLPS in this study, against the solubility scores obtained by SolupHred at pH 7.5 for all Y→S variants. We observed that the solubility of the variants was proportional to the number of Ser residues introduced in the sequence (R^2^ = 0.95) (Figure 7A) and thus that LLPS and solubility appear again to be correlated.

The most studied amino acid substitutions in LLPS-DRs are those leading to pathological aggregation. For example, mutations of the well-conserved Asp314 at the C-terminal LCD of hnRNPA1 to either Asn or Val divert the process of phase separation towards the formation of pathogenic amyloid fibrils, a process that is associated with ALS onset [38,39]. When studying the solubility curves of these two variants, a general decrease in maximum solubility was observed (Figure 7B), consistent with aggregation occurring inside liquid droplets, resulting in their rigidification and ultimately in the formation of stable amyloid assemblies.

Overall, tight solubility conditions are required for LLPS, and sequence modifications that alter this property, either towards more or less soluble states, impact the efficacy of the process.

### 3.7. pH-Dependent LLPS: Optimal Condition Evaluation

Finding the optimal conditions at which LLPS may occur for a given protein is not trivial; multiple variables must be assessed when studying this phenomenon. One recent study tried to establish a generic approach to study LLPS under near-native conditions [40]. In this work, the authors induced the formation of liquid condensates by the LCDs of hnRNPA2, TDP-43, NUP98, and ERD14 proteins by a pH jump from extreme pHs (3.0 or 11.0), where the proteins did not phase separate, to physiological pH where they found that liquid droplets were formed.

The solubility curves obtained by SolupHred for these LLPS-DRs (Figure 8) in the analyzed pH regime indicate that in the case of hnRNPA2, TDP-43, and ERD14, the condensation-promoting pH precisely maps within the range in which the LCDs manifest their maximum solubility. This is not the case with NUP98, for which the solubility at physiological pH is high, but not maximum, as this value is attained at very low pHs, where, in fact, LLPS was not observed. An inspection of the sequence of this protein indicates that it is highly cationic, and the high net charge of the region at acidic pH would compromise LLPS because of electrostatic repulsion. Decreasing this effect by moving toward neutral pHs would allow phase separation to occur in conditions where the solubility is still significantly high. An important corollary of this analysis is that LLPS does not ineludibly occur in conditions where the solubility is very low. It is important to note that according to our algorithm, the pI of a given protein does not necessarily coincide with the pH conditions at which the minimum solubility score is found, or on the other way around, that the pHs more distant from the pI are not always those at which the protein would exhibit maximum solubility. This results from considering simultaneously the impact of the pH in sequence hydrophobicity and net charge, not only this last factor. Indeed, for the above-mentioned proteins, the LLPS-DR pI is a poor predictor of conditions eliciting LLPS.

The discussed results still do not allow for the standardization of a method to predict or enhance LLPS just considering the solution pH independently. However, obtaining a pH-dependent solubility profile allows for an evaluation of the physicochemical parameters that may influence the formation of liquid condensates. As a general trend, naturally occurring LLPS-DRs can do it around neutral pH, which usually matches with intervals of significant solubility. Moving away from these regions destabilizes the multivalent-weak interactions required for LLPS, increasing the repulsive net charge or over-stabilizes them, permitting LLPS, but also the evolution of liquid droplets towards the formation of pathogenic aggregates.

## 4. Discussion

The intriguing phenomenon of LLPS has been attracting the attention of biologists and biophysicists in the last years. Many different observations of MLOs associated with a wide variety of IDPs have been reported in the literature [5,12,41]. However, large-scale analyses of the properties of these regions in context to their surrounding environment have remained elusive, despite the importance of extrinsic factors in modulating this process [6]. In this work, we have conducted a bioinformatics survey to investigate the effect of pH on the solubility and disorder of LLPS-DRs, allowing for better elucidation of the role of this solvent condition in the outcome of LLPS.

Our results indicate that LLPS-DRs present maximum solubility around neutral pH. This was initially intriguing, as previous studies suggested that APRs endorsing IDRs with lower local solubilities may be necessary to drive the formation of protein condensates [30,31]. Indeed, LLPS is a highly dynamic and reversible process that likely requires weaker interactions than those provided by highly aggregating patches. Therefore, LLPS-DRs need to display a significant degree of solubility to phase-separate into liquid droplets near physiological pH conditions. Intrinsic solubility is, however, an important determinant of autonomous phase separation, since psself IDRs exhibit, as a group, less soluble sequences than psother and DisProt IDRs, a property that is accompanied by a higher propensity to populate compact states. Still, for these interaction-prone sequences, LLPS occurs at pH values where they are predicted to exhibit significant solubility, a condition that would be compatible with their functioning and preclude the aggregation of these regions. This might be biologically important since psself IDRs’ solubility and disorder properties seem to have evolved to be more resistant to small pH perturbations. Our analysis provides a plausible answer to why some disordered regions phase-separate under specific pH conditions and others do not, although it is not intended to predict their behavior individually.

Mutations in psself LCDs can shift the equilibrium that sustains LLPS both to a more soluble or aggregated state, which may eventually lead to disease onset. In contrast, regions in the psother dataset need a partner to compensate for their lack of intrinsic condensation propensity. In these proteins, studying the contribution of mutations is much less straightforward, as pathogenicity may stem from poor interaction with its phase-separating partner or enhanced interaction with non-intended binders.

With this work, we aimed to start elucidating the role of pH in context-dependent phenomena such as LLPS and describe the general tendencies that arise from large-scale analysis. Given that many cellular processes and disease-associated pathways are sensible to cellular milieu fluctuations, we believe that by understanding the contribution of pH in both IDPs’ solubility and disorder at a large scale, we will be a step closer to understanding the impact of the environment in such processes.

## Figures and Tables

**Figure 1 biomolecules-12-00974-f001:**
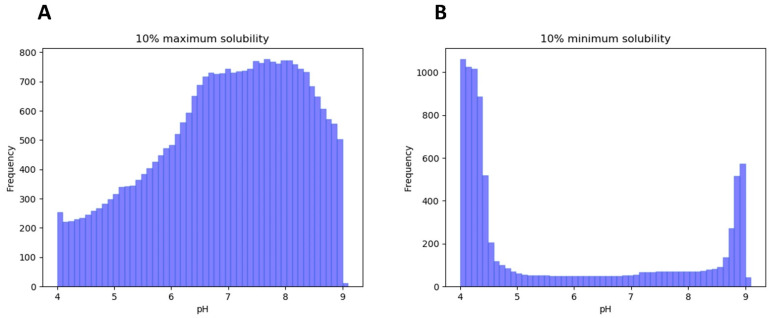
10% maximum (**A**) and 10% minimum (**B**) solubility distribution of LLPS-DRs in the range between pH 4 and 9 using step size 0.1. Maximum solubility is mostly attained around neutral pH (μ = 6.95), whereas minimum solubilities are found at the extremes of the pH interval, in more acidic or basic conditions.

**Figure 2 biomolecules-12-00974-f002:**
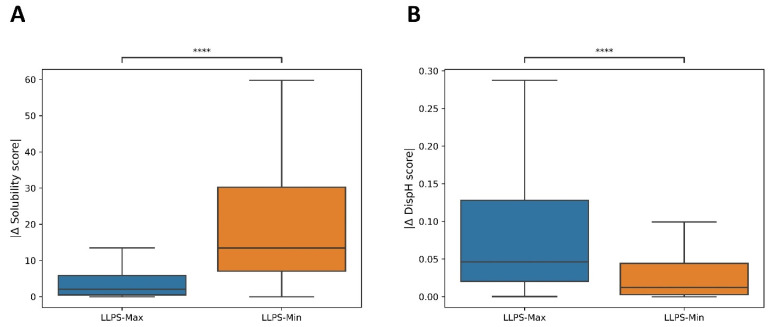
Absolute differences in LLPS-DR solubility (**A**) and disorder (**B**) scores between the predicted maximum or minimum value and the score at the pH where LLPS was described. pH conditions in LLPS buffers are closer to LLPS-DR maximum solubilities (*p* = 6.080 × 10^−31^) and minimum DispH scores (*p* = 3.905 × 10^−15^). Four asterisks (****) indicate *p* ≤ 0.0001.

**Figure 3 biomolecules-12-00974-f003:**
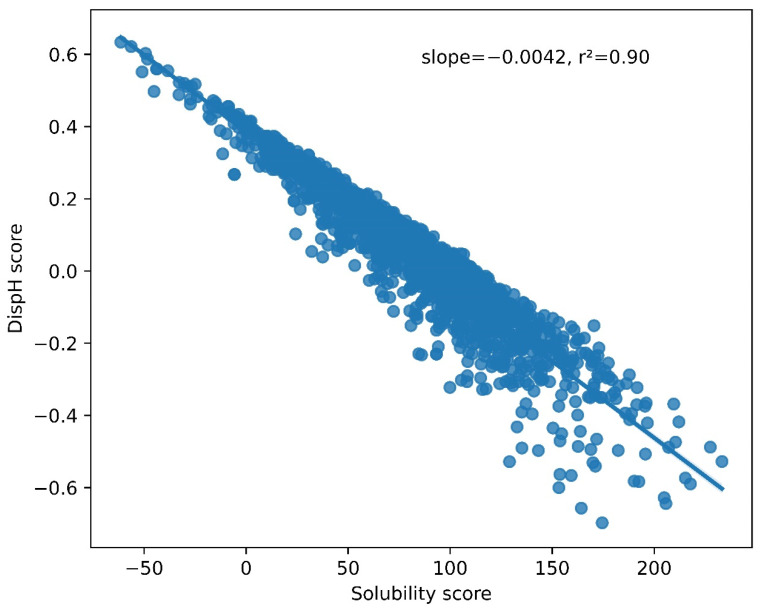
Correlation between LLPS-DR disorder and solubility scores at pH 7.0. A significant linear correlation can be observed (R^2^ = 0.90, *p* < 1^−10^). Higher solubility scores are associated with lower DispH scores (more disordered states).

**Figure 4 biomolecules-12-00974-f004:**
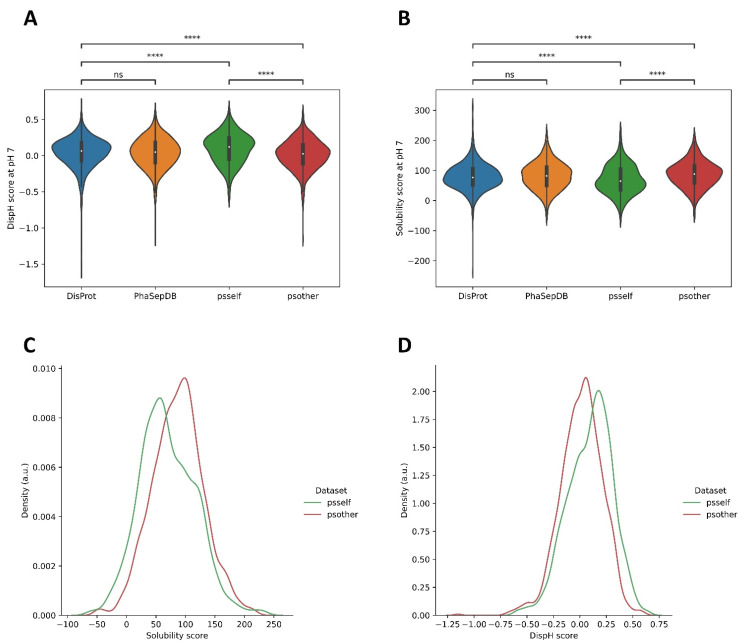
Comparison between solubility (**A**) and disorder (**B**) scores of DisProt, PhaSepDB, psself, and psother datasets at pH 7.0. Differences in disorder and solubility are non-significant between DisProt and PhaSepDB. However, when psself and psother are treated independently, major differences can be observed with DisProt and among these sub-sets. The psself dataset exhibits lower solubilities (**C**) and lower disorder (**D**) than psother sequences *. The density coordinate provides a representation of the number of observed sequences for each given score. * (**A**) Solubility: DisProt–psself (2.321 × 10^−05^); DisProt–psother (2.000 × 10^−07^); psself-psother (2.744 × 10^−12^). * (**B**) Disorder: DisProt–psself (3.282 × 10^−07^); DisProt–psother (2.902 × 10^−05^); psself-psother (4.457 × 10^−13^). Four asterisks (****) indicate *p* ≤ 0.0001. “ns” indicate *p* > 0.05.

**Figure 5 biomolecules-12-00974-f005:**
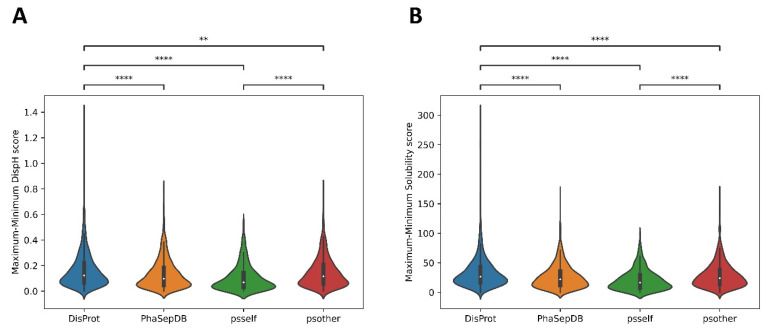
Difference between maximum and minimum solubility (**A**) and disorder (**B**) scores for DisProt, PhaSepDB, psself, and psother datasets. The DisProt dataset exhibits a wider dispersion, understood as the difference between the maximum and minimum values, in comparison with the PhaSepDB, psself, and psother datasets. In consonance with previous results, psother and psself datasets are significantly different *. * (**A**) Solubility: DisProt-PhaSepDB (1.526 × 10^−19^); DisProt–psself (2.064 × 10^−31^); DisProt–psother (5.508 × 10^−06^); psself-psother (2.009 × 10^−14^). * (**B**) Disorder: DisProt-PhaSepDB (4.496 × 10^−14^); DisProt–psself (3.325 × 10^−29^); DisProt–psother (5.526 × 10^−03^); psself-psother (8.586 × 10^−17^). Two asterisks (**) indicate *p* ≤ 0.01. Four asterisks (****) indicate *p* ≤ 0.0001.

**Figure 6 biomolecules-12-00974-f006:**
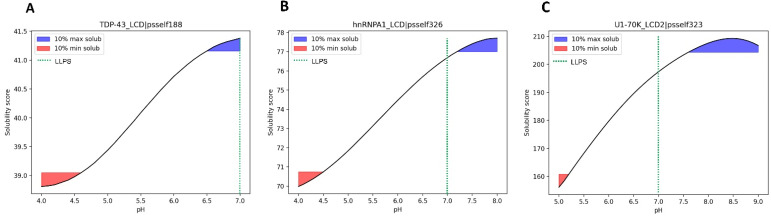
Predicted solubility curves at different pHs for the LCDs of TDP-43 (**A**), hnRNPA1 (**B**) and U1-70K (**C**). In all three cases, homotypic LLPS was observed at pH 7.0, a condition in which they display relatively high solubility scores.

**Figure 7 biomolecules-12-00974-f007:**
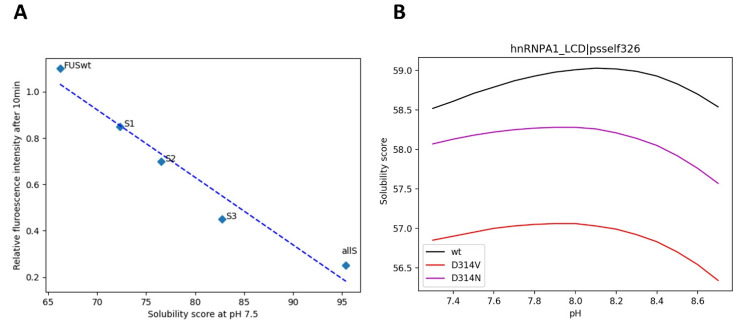
Predicted solubility scores for LLPS-impacting mutations in the LCD of FUS at pH 7.5 (**A**), and hnRNPA1 (**B**) in the 10% maximum solubility pH interval. Both solubilizing (**A**) and aggregating (**B**) mutations may alter LLPS equilibrium.

**Figure 8 biomolecules-12-00974-f008:**
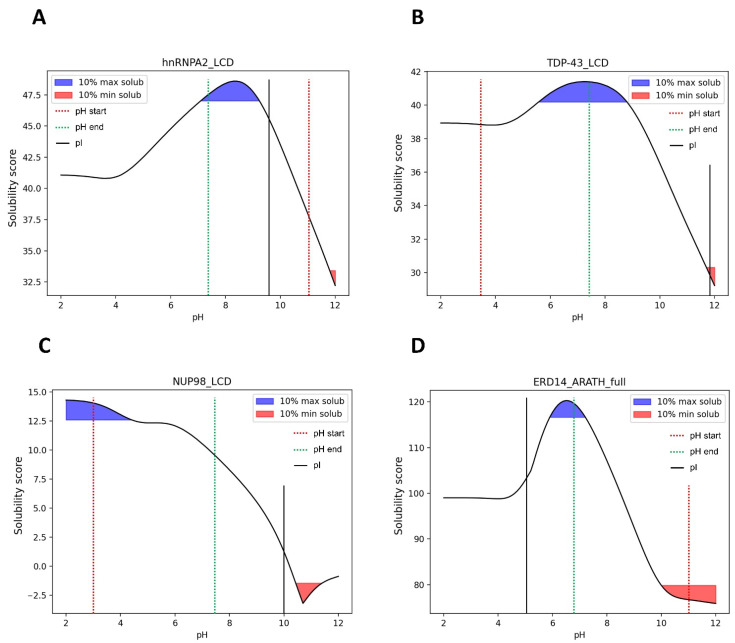
Predicted solubility curves in the pH jump interval for the LCDs of hnRNPA2 (**A**), TDP-43 (**B**), NUP98 (**C**) and ERD14 (**D**). The jump from an extreme pH (dotted red line) to near neutral pH (dotted green line) induces the formation of liquid droplets in conditions within -or close to- 10% maximum solubilities.

## Data Availability

Data supporting the results of this study can be accessed in the Appendix A of this article.

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
