# Peer review of "In-Silico Analysis of pH-Dependent Liquid-Liquid Phase Separation in Intrinsically Disordered Proteins"

_biomolecules, 2022, doi:10.3390/biom12070974_

Round 1

Reviewer 1 Report

Via a large-scale in-silico analysis of solubility and disorder of LLPS-IDRs, the authors investigated the role of pH in regulating LLPS. They found that LLPS-IDRs present maximum solubility and disorder around neutral pH where LLPS often occurs. They further found psself regions which undergo LLPS by themselves present less solubility and disorder at pH 7.0, and lower dispersion in solubility and disorder around neutral pH, when compared with psother regions which need partners to phase separate. Then they observed that LLPS and solubility are highly correlated, and mutations changing the solubility of LLPS-DRs may alter the LLPS equilibrium. Finally, they suggested that pH-dependent solubility is a good predictor of conditions compatible with LLPS.

Overall, this paper provides a potential explanation of how the solution pH influences LLPS of LLPS-DRs. The model sounds interesting, however, the manuscript suffers from several weaknesses leaving their main conclusions only suggestive. My concerns are listed bellow:

 1.     The authors chose 10% maximum/minimum solubilities as the most representative variables for the analysis (Fig1). What’s the reason to select this number? Base on Fig1, they suggested that LLPS-DRs present maximum solubility around neutral pH. However, the predicted solubility scores of hnRNPA1 and U1-70K are not maximum at pH 7.0, where both proteins exhibit the strongest LLPS property. Actually, pH 7.0 resides near the middle between the pH values with 10% max and min solubilities (Fig.6). Another case is NUP98, which presents the max solubility at very low pH (Fig. 8C). It is not clear how many LLPS-DRs indeed exhibit a highly correlated LLPS property and solubility around neutral pH.   

2.     Based on the statistics in Fig4A and B, the authors claimed psself proteins are less soluble and disordered than psother and disordered proteins at pH 7.0. However, all the error bars in these figures are very large, reflecting dramatic score variation among different proteins. Thus, the conclusion may be not applicable for a substantial portion of proteins undergoing LLPS.

3.     Page 6 line 200, it would be better to change "in physiological pH" to "in the physiological pH range".

4.     Page 6 line 202, the “selected pH range” should be specified.

5.     For Fig 4 C and D, how the coordinate “density” was obtained should be described in the figure legends or Methods.

Reviewer 2 Report

The manuscript submitted by C. Pintado-Grima, O. Barcenas, and S. Ventura describes the influence of pH on liquid-liquid phase separation of disordered proteins. The most important discovery is probably the following: “psself” and “psothers” are different.

The manuscript is well written, perhaps a bit verbose.

I have only one concern: a threshold of 20 aa was selected (section 2.1). It is not too short?

Two other minor problems: (i) in section 2.2 the Mann-Whitney-Wilcoxon two-sided test and the Wald test are mentioned; perhaps they might be described to the readers of biomolecules, maybe in the Supplementary Material. (ii) In Figures 6-8, the vertical axis indicates the “Solubility” without units (this is just a typo).

Round 2

Reviewer 1 Report

My past concerns have been addressed adequately. The revised paper describes an interesting study that should be of general interest. I recommend that it is accepted without further revisions.